# The Utility of Data Transformation for Alignment, De Novo Assembly and Classification of Short Read Virus Sequences

**DOI:** 10.3390/v11050394

**Published:** 2019-04-26

**Authors:** Avraam Tapinos, Bede Constantinides, My V. T. Phan, Samaneh Kouchaki, Matthew Cotten, David L. Robertson

**Affiliations:** 1School of Biological Sciences, The University of Manchester, Manchester M13 9PT, UK; bede.constantinides@manchester.ac.uk (B.C.); samaneh.kouchaki@eng.ox.ac.uk (S.K.); david.l.robertson@glasgow.ac.uk (D.L.R.); 2Modernising Medical Microbiology Consortium, Nuffield Department of Clinical Medicine, John Radcliffe Hospital, University of Oxford, Oxford OX3 9DU, UK; 3Department of Viroscience, Erasmus Medical Centre, Doctor Molewaterplein 40, 3015 GD Rotterdam, The Netherlands; v.t.m.phan@erasmusmc.nl (M.V.T.P.); mlcotten13@gmail.com (M.C.); 4Department of Engineering Science, Institute of Biomedical Engineering, University of Oxford, Oxford OX3 7DQ, UK; 5MRC-University of Glasgow Centre for Virus Research, Glasgow G61 1QH, UK; 6MRC/UVRI & LSHTM Uganda Research Unit Entebbe, P.O. Box 49 Entebbe, Uganda

**Keywords:** alignment, assembly, taxonomic classification, time series, data transformation, DWT, DFT, PAA, data compression, compressive genomics

## Abstract

Advances in DNA sequencing technology are facilitating genomic analyses of unprecedented scope and scale, widening the gap between our abilities to generate and fully exploit biological sequence data. Comparable analytical challenges are encountered in other data-intensive fields involving sequential data, such as signal processing, in which dimensionality reduction (i.e., compression) methods are routinely used to lessen the computational burden of analyses. In this work, we explored the application of dimensionality reduction methods to numerically represent high-throughput sequence data for three important biological applications of virus sequence data: reference-based mapping, short sequence classification and de novo assembly. Leveraging highly compressed sequence transformations to accelerate sequence comparison, our approach yielded comparable accuracy to existing approaches, further demonstrating its suitability for sequences originating from diverse virus populations. We assessed the application of our methodology using both synthetic and real viral pathogen sequences. Our results show that the use of highly compressed sequence approximations can provide accurate results, with analytical performance retained and even enhanced through appropriate dimensionality reduction of sequence data.

## 1. Introduction

Next-generation sequencing (NGS) enables massively parallel determination of nucleotide order within genetic material, making it possible to rapidly sequence the genomes of individuals, populations and metagenomic samples [1,2,3,4,5]. However, the sequences generated by these instruments are almost always considerably shorter in length than the genomic regions studied. Genomic analyses often begin with the process of sequence assembly, where sequence fragments (reads) are reconstructed into the larger sequences from which they originated. Computational methods play a vital role in the assembly of short reads, and a variety of assemblers and related tools have been developed in tandem with emerging sequencing platforms [6]. All subsequent analyses and investigations depend upon the quality, accuracy and speed of this crucial sequence assembly process. 

There are many computational methods to generate consensus sequences representing the genomes of species in a sample. Such approaches include seed-and-extend alignment methods using suffix array derivatives, such as the Burrows-Wheeler Transform (BWT) for aligning short reads informed by a known reference sequence [7,8], graph-based methods employing Overlap Layout Consensus (OLC) [9,10] and de Bruijn graphs of *k*-mers [11,12,13] for reference-free de novo sequence assembly. However, for sequencing projects to characterise genetic variation within populations (deep sequencing), metagenomics and pathogen discovery, the effectiveness of the aforementioned approaches varies considerably [14].

Samples with mixed viral infections, especially those comprising divergent variants, present a number of analytical and computational problems. The use of a reference sequence, even the use of a data specific generated sequence, can lead to valuable read information being discarded during the alignment process [15]. On the other hand, while de novo approaches require little a priori knowledge of target sequence composition, the methods are computationally intensive, and their performance scales poorly with datasets of increasing size [9]. Aggressive heuristics must be employed, to traverse graphs and deal with mismatches, reduce the running time of de novo assemblers, which, in turn, can compromise assembly quality. Indexing structures such as the BWT and its relatives are widely used to reduce the burden of pairwise sequence comparison, for both reference-based mapping and de novo assembly. However, they cannot process mismatches within reads, necessitating the use of computationally expensive heuristics to establish relationships between divergent sequences. Increasing sequence length further affects the performance of these approaches [16].

A major challenge in working with NGS data from metagenomic studies is the high levels of diversity present, particularly for the virus genetic material. Also, the number of sequences generated challenge many computational systems for a feasible working solution in terms of time and the computational resources typically available in biological laboratories. For biologists working on outbreak responses or pathogen discovery, both the accuracy of the assembly results and the speed of sequence analyses (e.g., assembly, alignment and pathogen classification) are crucial for crisis response and management. The ability to run analyses in the field on portable computer systems without internet connectivity is also important. Here, we explore the utility of data transform methods to extract major features from viral NGS sequence data and use the features to analyse data in a lower dimensional space.

Similar analytical challenges involving high dimensional sequential data are encountered in other data-intensive fields, such as signal and image processing, and time series analysis, where data transforms and approximation techniques are used for data dimensionality reduction. Data transform/approximation techniques include the discrete Fourier transform (DFT) [17], the discrete wavelet transform (DWT) [18,19] and piece-wise aggregate approximation (PAA) [20,21]. The DFT or DWT are used to transform data to their frequency domains, allowing feature extraction [22], and PAA is used as a data approximation approach. In data-intensive fields, data transformations/approximations are commonly used as dimensionality reduction approaches for obtaining fast approximate solutions for a given problem. Due to the ordered nature of genetic data, many of these transformation approaches can be applied to sequences of nucleotides [23] or amino acids [24]. An example of a successful implementation of a Fourier transform in computational biology is the multiple sequence alignment based on fast Fourier transform alignment algorithm MAFFT [25] where the physiochemical properties of amino acids are used to represent sequences for fast matching of homologous sequence regions for alignment. Since most transformation approaches are suitable only for numerical sequences, the strings of letters representing genetic sequences must be mapped into numerical space using a numerical sequence representation method [26]. 

In addition to the DFT, the DWT and PAA, suitable methods for measuring the pairwise similarity of sequential data or transformations include the Lp-norms [27], dynamic time warping (DTW) [28], longest common subsequence (LCS) [29], and alignment approaches, such as the Needleman-Wunsch and Smith-Waterman algorithms. Euclidean distance is arguably the most widely used Lp-norm method for sequential data comparison but can only be used on sequences of the same length. Furthermore, Lp-norm methods do not accommodate shifts in the *x*-axis (time or position) and are thus limited in their ability to identify similar features within offset data. Elastic similarity/dissimilarity methods, such as LCS, unbounded DTW and various alignment algorithms, permit comparison of data with different dimensions and tolerate shifts in the *x*-axis. These properties of elastic similarity methods can be very useful in the analysis of speech signals, for example, but can be computationally expensive [30]. Several approaches have been proposed to permit fast searching with DTW, including the introduction of global constraints (wrapping path) or the use of lower bounding techniques, such as LB_keogh [28]. 

While pairwise comparison methods may be used for clustering, classification and similarity searches, they are very time consuming for large datasets (*O(n^2^)* time complexity). Indexing structures, such as the *R**-tree, *KD*-tree, *VP*-tree and *MVP*-tree have significantly lower time complexity (*O(n log(n))*) for similarity search [31] and are more appropriate for efficient analysis of large datasets. The *R**-tree [32,33] and *KD*-tree [34] indexing structures are very accurate for low dimensional datasets. However, their performance deteriorates significantly in high dimensional space [31], a phenomenon known as the ‘curse of dimensionality’ [35,36]. Metric trees, such as the *VP*-tree [37] and *MVP*-tree [38], are less prone to this limitation. Metric space indexing structures make use of geometric properties for partitioning data and work efficiently on both low and high dimensional data [39]. The curse of dimensionality can be further mitigated using data approximations, such as the DFT, the DWT and the PAA, to partition a dataset in an approximated space without loss of generality [21].

Here, we investigate the performance of three established dimensionality reduction techniques on three common analysis tasks involving viral short read sequence data: classification, reference-based mapping/alignment and de novo assembly. We benchmarked the accuracy of our proposed methodology against existing tools, and demonstrate the applicability of time series and signal processing data mining techniques for the analysis of viral NGS data.

## 2. Materials and Methods 

### 2.1. Symbolic to Numeric Sequence Representations

Various numeric sequence representation methods can be used for symbolising a nucleotide sequence to a numerical space (see 51). Depending on the chosen numerical representation, each nucleotide is associated with a specific numerical value or vector. The specific values are assigned to the position of each nucleotide indicating the presence of a nucleotide at each sequence position (Equation 1). *R_i_* is the indicator for a specific nucleotide in the *i^th^* position of the sequence *S* with a length of *n* nucleotides. Values *v*_1_…*v*_5_ correspond to the numerical value or numerical vector associated with each nucleotide.
(1)$$R=\{\begin{array}{c} v_{1}\;if\;i=A\\ v_{2}\;if\;i=T\\ v_{3}\;if\;i=C\\ v_{4}\;if\;i=G\\ v_{5}\;otherwise \end{array}\forall i\in S$$

Methods, such as the electron-ion interaction pseudopotentials (EIIP) [40] and the atomic representation approach [41], aim to mimic the biochemical properties of nucleic acids but introduce some mathematical bias that does not exist in reality [26]. Other methods, like the Voss indicator [42] and the Tetrahedron approach, do not introduce internucleotide mathematical bias, meaning the pairwise distances between each non-identical transformed nucleotide are the same (for example, the distance between A and T is equal to the distances between A and C as well as A and G). Furthermore, the cumulative sum of a numerical representation *R* can be used to indicate the trajectory of a sequence in nucleotide space. Table 1 indicates the values used for different representation methods [26].

### 2.2. Sequence Transformation

Effective methods for transforming/approximating sequential data should: (i) accurately transform/approximate data without loss of useful information, (ii) have low computational overheads, (iii) facilitate rapid comparison of data and (iv) provide lower bounding—where the distance between data representations is always less than or equal to that of the original data—precluding false negative results [43]. The lower bounding property guarantees that if two data points are nearby in their original space, they will remain so in their transformed/approximate space. We employ the DFT and the DWT transformation methods and the PAA approximation method as they satisfy the above requirements, and these are widely used for analysing discrete signals [44] and can be used to transform/approximate nucleotide sequence numerical representations to different levels of resolution, permitting reduced dimensionality sequence analysis.

Figure 1A illustrates an example of the DFT and DWT transformations and PAA approximation of a short nucleotide sequence. The DFT and the fast Fourier transform (FFT) convert data from their original domain into the frequency domain. In principle, the DFT decomposes a numerically represented nucleotide sequence with *n* positions (dimensions) into a series of *n* frequency components ordered by their frequency. A subset of the resulting Fourier frequencies are used to approximate the original sequence in a lower dimensional space [17], and the tradeoff between analytical speed and accuracy can be varied according to the number of frequencies considered [45].

The DWT transforms data into the time-frequency domain, capturing both frequency and temporal location information [18,46,47], in contrast to DFT, which only provides frequency information. DWT is a set of averaging and differencing functions that may be used recursively to represent sequential data at different resolutions, and each resolution can be used as an approximation of the original data. Figure 1B depicts DWT transformations of a short nucleotide sequence.

In PAA, a numerical sequence is divided into *n* equally sized windows, the mean values of which together form a compressed sequence representation [20,21]. The selection of *n* determines the resolution of the compressed or approximate representation. While PAA is faster and easier to implement than the DFT and the DWT, unlike these two methods, PAA is irreversible, meaning that the original sequence cannot be recovered from the approximation. Figure 1C depicts an example of the PAA transformations of a short nucleotide sequence.

### 2.3. Similarity Search Approaches for Sequential Data

Here, we adopt the Euclidian distance and *VP*-tree index to perform a fast *k*-nearest neighbour (*k*-NN) similarity search for aligning the reads to a reference genome.

In a *VP*-tree indexing structure, data is segregated using the distance between data points, thus implementing data partitioning in a metric space. A data point to use as a vantage point is selected (either randomly or by applying some heuristic to find and use the furthest point in the dataset [37]), and the rest of the data points are partitioned into two nodes based on their distance to that point. Data found to be closer to the vantage point than a given threshold (the median distance between all the data points and the vantage point) are assigned to the same node, and the rest of the data points to a different node. This function is repeated recursively in order to complete the partitioning process. The resulting indexing structure can then be used for fast identification of a *k*-nearest neighbour (*k*-NN) search. A *k*-NN-search returns the data points that are closest to a query *q*. Initially, the distance between the query *q* and the vantage point in the top node is calculated. If the distance between *q* and the vantage point satisfies a set of given conditions (the distance is smaller or larger than a given threshold – this threshold being the median distance between the vantage point and other data points within the node), a decision is made to visit either one or both of the child nodes. This process is repeated until the entire tree has been traversed. The *k* data points—in this case, reads—found closest to our query are the *k*-nearest neighbours to the query *q*.

### 2.4. Proposed Short Reads Processing Methodology

Our methodology for taxonomic classification, reference-based mapping and de novo assembly of short reads used time series and digital signal processing data transformation techniques. Figure 2 illustrates the fundamental concept of our approach. The short reads and reference genomes are mapped to a numerical space using an appropriate method from Table 1. Subsequently, lower dimensional approximations were generated for all data using the appropriate data transformation method, such as DFT, DWT and PAA. A *VP*-tree was constructed to allow fast data comparison. Depending on the application, the *VP*-tree was constructed either by using *k*-mer transformations obtained from the reference genomes or by using the short reads’ transformations. Consequently, the best matches for our short reads’ transformations were identified using a *k*-NN search approach on the *VP*-tree. As a final step, the results obtained from the *k*-NN search were re-evaluated in the original space to remove potential false positive results.

### 2.5. Data

The implementations of our proposed methodologies were assessed with both simulated and real virus datasets. The simulated datasets were generated using CuReSim [48] and WGSIM (https://github.com/lh3/wgsim). Simulated data included information, such as the reference genome used, the alignment position and alignment direction, for each read, enabling rigorous evaluation of the proposed techniques. We used two simulators to examine our approach in a variety of use cases. CuReSim is highly customisable, allowing the user to control the type of variation (insertion, deletion and substitution) to simulate. WGSIM can simulate genomes with uniform insertion, deletion and substitution variation.

CuReSim was used to generate 16 HIV-1 HXB2 simulated datasets with different levels and types of variation. WGSIM was used to generate 4 mixed virus datasets with different levels of variation. Each simulation contained 200,000 reads generated using 5 Norovirus, 5 Ebola virus and 5 Respiratory syncytial virus (RSV) genomes, with various types and extents of simulated variation. HXB2 and simulated mixed virus datasets and corresponding reference genomes used to simulate them are deposited on GitHub (https://github.com/Avramis/Supporting-data/tree/master/Simulated%20Data). Table 2 contains detailed information about the simulated datasets.

Furthermore, 15 publicly available real virus datasets were used for the evaluation of our methodology. The real datasets comprise 5 Norovirus, 5 Ebola virus and 5 human respiratory syncytial virus (RSV) short read datasets. Norovirus NGS datasets (ERR225628, ERR225629, ERR225631, ERR225632, ERR225633) were generated from diarrhoeal patients in Vietnam [49]. Group A rotavirus datasets were obtained from human and pig samples from Vietnam [50]. Human coronavirus NL63 datasets were obtained from Kenya [51]. The Ebola virus datasets (SRR3107337, SRR3107338, SRR3107340, SRR3107342, SRR3107343) were retrieved from the bioproject PRJNA309162, generated during the outbreaks in West Africa in 2013–2016 [52]. The human respiratory syncytial virus (RSV) datasets (ERR303259, ERR303260, ERR303261, ERR303262, ERR303263) [53] were generated from humans in Kenya. All 15 datasets are publicly available. The accession numbers of Sequence Read Archive (SRA) and European Nucleotide Archive (ENA) can be found in Table 3.

The HIV-1 HXB2 genome (K03455) was used as a reference index to align and/or run the taxonomic classification analysis for the HXB2 simulated dataset. The Norovirus genome KM198509, the Ebola virus genome KM034562 and the RSV genome KP317934 were used as a reference index to align and/or run the taxonomic classification analysis for the mixed virus datasets. The Norovirus genome KM198509 was used to run the taxonomic classification analysis on the real Norovirus datasets, the Ebola virus genome KM034562 was used to run the taxonomic classification analysis on the real Ebola datasets and the RSV genome KP317934 was used to perform the taxonomic classification analysis on the real RSV datasets. All reference genomes used in this study are available from the NCBI (https://www.ncbi.nlm.nih.gov/genome), and accession numbers can be found in Table 4.

### 2.6. Classification and Alignment Evaluation

The accuracy of a classification and an alignment tool can be quantified in terms of the *F*-measure [48], a balanced measure of precision and recall, with precision = true positive/true positive + false positive, recall = true positive/true positive + false negative and the *F*-measure = 2 × (precision *x* recall)/(precision + recall) [48]. In the case of simulated data, information concerning the position of the read on the reference and alignment direction can be used to establish the correctness of alignment, and thereby provide a more informative *F*-measure score. Unclassified reads are considered a false negative result. Any reported match to the correct region of the genome in the correct direction is considered a true positive result. However, if the alignment position or direction information is unavailable, the *F*-measure can be calculated from the number of hits reported for a read, or the absence of a hit. Again, unclassified reads are considered false negative results, and classified reads are considered true positive results. In the case of mixed genome data, the *F*-measure score can be calculated by taking into consideration the number of hits that are reported for a read, as well as if a read is assigned to a reference genome from the same family. If a read is assigned to a genome from a different virus family, it is considered a false positive result, while unclassified reads are considered a false negative result.

## 3. Results

### 3.1. Classification by Numbers (CBN)

For the taxonomic classification analysis, a classification tool was implemented in *C++* (https://github.com/Avramis/ClassificationByNumbers). The implementation was developed to evaluate our methodology but was not optimised for speed. Users might specify parameters, such as the representation method, transformation method, search stringency and the *k*-mer length. A *VP*-tree indexing structure classified reads using a given set of genomic references. *VP*-tree construction began with the extraction of all unique *k*-mers, of a user-specified length *k*, from the set of supplied reference genomes. Each unique *k*-mer was represented in numerical sequence and then transformed into a lower dimensional space. The transformed data were then used to generate the *VP*-tree indexing structure. Subsequently, each short read from a query set was converted into numerical space, transformed to a lower dimensional space and evaluated against the *VP*-tree. The approximate solution arising from this was then evaluated using the original data to identify false positive matches. The CBN algorithm generated two output files. The first output was a text file providing detailed information on all of the classification matches generated for each read, including the reference name, the direction in which the query read was aligned to the reference, the start and end position of the query on the reference, the alignment score, the CIGAR string describing how the read aligns with the reference and the actual alignment of the query read on the reference genome. The second tabular output file provided a brief overview of the alignment. Each line contained the name of the read, the number of classifications generated for that particular read, the highest classification score obtained, the name of the reference, which provided the highest classification score, the alignment direction and starting position on the reference.

The CBN tool was evaluated against NCBI-BLAST 2.8.1 BLASTn [54] and Kaiju 1.6.3 [55] classifier tools. BLASTn performs the analysis in nucleotide space, whereas Kaiju translates nucleotide sequences from every possible reading frame and performs the analysis in protein sequence space. Figure 3, Figure 4 and Figure 5 illustrate the results of the classification evaluation process. Both BLASTn and Kaiju were evaluated using their default parameters. CBN was evaluated using *k*-mers of 100, 150, 200, 250 and 300 for the HXB2 simulated reads and 50, 100 and 150 for the mixed virus and real datasets. For the DFT and PAA methods, we evaluated the use of transformation/approximations with 2, 4, 6, 8, 10 and 12 Fourier frequencies or PAA coefficients, respectively. For the DWT variant, we tested the cases of 2, 4, 8, 16 and 32 wavelets.

Figure 3 shows the results obtained from the classification process on HIV- 1 HXB2 data. Figure 4 illustrates the results of the mixed virus datasets. Figure 5 illustrates the results obtained from the real data. For taxonomic classification of HIV-1 HXB2 simulated reads, where the short reads were classified against the genome used to generate them, Kaiju reported the highest accuracy scores. CBN outperformed BLASTn in most cases, falling behind in terms of accuracy only on datasets with high variation rates. For the mixed virus simulated datasets, where reads were classified against species strains related to those used to generate reads, BLASTn correctly assigned the most species, followed closely by CBN and finally Kaiju. In the evaluation of the tools on the real data, where reads were classified using a publicly available species-specific reference sequence, CBN generated more accurate results than other tools, followed by Kaiju and BLASTn.

### 3.2. Alignment by Numbers (ALBN)

To test the applicability of sequential data transformations and feature selection for read alignment, we implemented a prototype *k*-NN read aligner (Figure 6) in *C++* (available at https://github.com/Avramis/Alignment_by_numbers). As with the CBN classification analysis, the ALBN code was not optimised for speed. Users might specify parameters, such as the representation method, transformation method, search stringency and the *k*-mer length used for seeding alignments. The algorithm’s output was used to construct gapped alignments in the widely used Sequence Alignment/Map (SAM) file format.

The ALBN tool was evaluated against a set of well-established, widely used, state-of-the-art tools, such as Bowtie2 (version 2.3.1) [56], BWA-MEM (version 0.7.16) [7], GraphMap (version 0.5.2) [57] and Segmehl (version 0.3.4) [58]. Existing state-of-the-art tools were evaluated with default settings. ALBN was evaluated using *k*-mer lengths of 100, 150, 200, 250 and 300 for the HXB2 simulated reads and 50, 100 and 150 for the mixed virus and real datasets. For the DFT and PAA variants, we evaluated the use of transformation/approximations with 2, 4, 6, 8, 10 and 12 frequencies and PAA coefficients accordingly. For the DWT variant, we tested the cases of 2, 4, 8, 16 and 32 wavelets.

Each aligner’s accuracy was quantified in terms of the *F*-measure [48]. CuReSim provides information, such as the simulated read’s origin on the reference genome and its alignment direction, enabling evaluation of each aligner’s output and calculation of alignment accuracy in terms of the *F*-measure. For mixed virus datasets, tool performance was evaluated in terms of ability to match and align reads to the appropriate virus reference genome. For the real data, *F*-measures were calculated according to the number of reads aligned to the given genome or otherwise.

Figure 7, Figure 8 and Figure 9 illustrate the *F*-measures obtained by evaluating alignments from each aligner. Figure 7 illustrates alignment performance for each of the 16 datasets simulated using the K03455 HIV-1 HXB2 reference genome. Figure 8 illustrates the alignment performance for virus reads simulated with Norovirus genome KM198509.1, Ebola genome KM034562.1 and the RSV genome KP317934.1. Figure 9i–iii illustrate alignment performance (*F*-measure) for alignments of real Norovirus, Ebola virus and RSV sequences against the same reference genomes as those used for simulation.

ALBN provided accurate results in all scenarios tested. Regarding the HIV-1 HXB2 data, where short reads were aligned to the genome used to generate them, ALBN provided the most accurate results in all 16 cases, followed by Bowtie2. This was also the case for the mixed virus datasets, where reads were aligned to reference strains related to those used to generate the dataset. In both cases, GraphMap and BWA-MEM were third and fourth in terms of accuracy, respectively. ALBN also generated the most accurate alignment results using real data, where reads were aligned to species-specific reference genomes.

### 3.3. De novo Assembly by Numbers

Lastly, to test the applicability of this approach to the de novo assembly of short reads, we implemented assembly by numbers (ASBN), a prototype algorithm for all-against-all *k*-mer comparison, using data transformations/approximation. Note, preliminary results have been presented as a conference paper [59]. Figure 10 illustrates the main concept of our de novo assembly approach. For the ASBN tool, reads are represented as numerical sequences using an appropriate numerical representation method (Table 1). Here, we used the tetrahedron numerical representation. Every *k*-mer of each numerically represented read was identified and transformed to lower dimensional space using the chosen transformation method. All *k*-mers’ transformations were used to build a *VP*-tree, to allow for fast data comparison. Afterwards, all *k*-mers were compared to the rest of the data using the *VP*-tree index. Information from the data comparison was used to construct a weighted graph similar to that shown in Figure 10A. The shortest path on the weighted graph was identified with a breadth-first search (BFS) (Figure 10B). Reads overlaps were used to generate an OLC alignment of short reads (Figure 10C).

The ASBN assembler was compared with Megahit (version 1.1.3) [60] and SPAdes (version 3.13.0) [61] de novo assemblers on the HIV-1 HXB2, and mixed virus simulated datasets accordingly. Megahit, SPAdes, BLASTn and Kaiju were evaluated using default parameters. ASBN was evaluated using *k*-mer lengths 100, 150, 200, 250 and 300 for the HXB2 simulated reads and 50, 100 and 150 for the mixed virus datasets. For the DFT and PAA variants, we evaluated the use of transformation/approximations with 2, 4, 6, 8, 10 and 12 frequencies and PAA coefficients accordingly. For the DWT variant, we tested the cases of 2, 4, 8, 16 and 32 wavelets.

The derived contigs from each assembler were evaluated against the reference genomes used to generate the data simulations with BLASTn [54]. From the BLASTn output, information about the contigs’ alignment position on the genome and the length of the alignment were obtained. Subsequently, a measure of assembly contiguity and the sum of gaps/mismatches were calculated and plotted on an X-Y matrix (similar to Figure 11 and Figure 12) with *x* being the total coverage of the genomes generated and *y* being the total number of gaps in the coverage. A perfect assembly would have *x* = full genome length and *y* = 0, indicating that the contig is identical to the genome in terms of length and nucleotide composition. For the HIV-1 HXB2 datasets, the contigs were evaluated against the K03455 genome, and the contigs obtained from the mixed virus datasets were evaluated against the 15 different genomes: KM198529, KM198528, KM198511, KM198500, KM198486, KU296608, KU296553, KU296549, KU296528, KU296416, KP317952, KP317946, KP317934, KP317923 and KP317922.

Figure 11 illustrates the assembly results of SPAdes, Megahit and all three variants of ASBN on the 16 simulated HIV-1 HXB2 datasets. Figure 12 illustrates the assembly results on the mixed virus simulated databases. Although ASBN processes data and assembles short reads in a lower dimensional space, it nevertheless generated contigs that collectively cover the expected genome length and provided comparable results to both existing state of the art de novo assemblers tested in this experiment (Figure 11 and Figure 12). In all cases, ASBN generated contigs spanning the whole genomes of their respective viral species.

## 4. Discussion

Although well-established data compression methods for reversible compression of one-dimensional and multivariate signals, images, text and binary exist [62,63,64], there are very few examples of their application to biological sequence data. We have developed algorithms incorporating signal compression methods for three common biological sequence analysis problems: classification, alignment and de novo assembly of NGS short read virus data. Our results in Figure 3, Figure 4, Figure 5, Figure 6, Figure 7, Figure 8, Figure 9, Figure 10, Figure 11 and Figure 12 show that this approach permits accurate classification of de novo assembly and reference alignment in spite of high rates of sequence variation or the use of a divergent reference genome. Data approximation/summarisation techniques, such as the DFT, the DWT and the PAA, can be used to extract major features of sequence data and to suppress noise or low-level variation. This allows sequence comparison exploiting the major characteristics of the data, thus enabling the identification of similarities that might otherwise be concealed by minor variation or sequencing error/noise.

Collectively, our results demonstrate that complete nucleotide-level sequence resolution is not a prerequisite of accurate sequence analysis and that analytical performance can be preserved and even enhanced through appropriate dimensionality reduction (compression) of sequences. While our implementations use *k*-mers, the nature of the transformation/compression methods used shows that optimal *k*-mer length selection is far less important than the conventional exact *k*-mer matching methods. The inherent error tolerance of the approach also permits the use of longer *k*-mers than typically used in conventional sequence comparison algorithms, reducing the computational burden of pairwise comparison, and thus, in de novo assembly specifically, the complexity of building and searching an assembly graph.

Efficient mining of terabase-scale biological sequence datasets requires looking beyond substring-indexing algorithms [65] towards more versatile methods of compression for both data storage and analysis. The use of probabilistic data structures can considerably reduce the computer memory required for in-memory sequence lookups at the expense of a few false positives, and Bloom filters and related data structures have seen broad application in *k*-mer centric tasks, such as error correction [66], in silico read normalisation [67] and de novo assembly [68,69]. However, while these hash-based approaches perform well on datasets with high sequence redundancy, for large datasets with many distinct *k*-mers, large amounts of memory are still necessary [67]. Lower bounding transformations and approximation methods (such as the DFT, the DWT and PAA) can exhibit the same attractive one-sided error offered by these probabilistic data structures, but instead of hash tables, use concrete and reusable sequence representations.

Furthermore, transformations allow compression of standalone sequence composition, enabling flexible reduction of sequence resolution according to analytical requirements, so that redundant sequence precision need not hinder analysis. While the problem of read alignment to a known reference sequence is largely considered solved, assembly of large genomes remains a formidable problem in computing. Moreover, consideration of the metagenomic composition of mixed biological samples, as demonstrated, further extends the scope and scale of the assembly problem beyond what is tractable using conventional sequence comparison approaches. By implementing a reference-based aligner and de novo assembler, we have demonstrated that using compressed numerical representations offers a versatile approach for reconstructing genomes and metagenomes sequenced with short reads.

Emerging long read sequencing technologies bring new challenges for sequence data analysis. Whilst the error rate of Oxford Nanopore sequencing platform, for example, has decreased considerably since the technology’s introduction [70,71], the relatively high error rate still limits the scope of downstream analyses [72]. Efficient algorithmic approaches are needed to (1) identify sequence identity/infer homology in spite of abundant insertion/deletion errors associated with the platform, which are problematic for approaches dependent on exact subsequence matching and (2) to overcome issues relating to high data dimensionality and the curse of dimensionality [73]. Both in terms of the raw electric current traces generated by DNA translocation through a nanopore and the corresponding base-called sequences, the resemblance between long reads and time series data from other fields is striking, such that the various transformations/approximations we have implemented will be directly applicable.

In conclusion, nucleotide sequences may be effectively represented as numerical series, enabling the application of existing analytical methods from a variety of mathematical and engineering fields for the purposes of sequence alignment and assembly. By applying established signal decomposition methods, compressed representations of nucleotide sequences can be created, permitting reductions in the spatiotemporal complexity of their analysis, without necessarily compromising analytical accuracy.

## Figures and Tables

**Figure 1 viruses-11-00394-f001:**
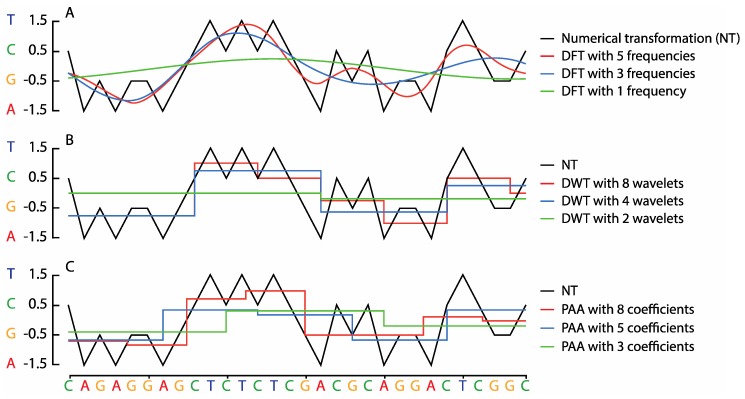
A numerically represented DNA sequence transformed at various levels of spatial resolution using the discrete Fourier transform (DFT) of the whole sequence (**A**), the Haar discrete wavelet transform (DWT) (**B**) and piecewise aggregate approximation (PAA) (**C**). A 30 nucleotide sequence (*x*-axis) is represented as a numerical sequence (black lines) using the real number representation method (*y*-axis where T = 1.5, C = 0.5, G = −0.5 and A = −1.5) for DFT approximations of the sequence with 5 (red), 3 (blue) and 1 (green) Fourier frequencies (**A**); DWT approximations of the same sequence with 8 level wavelets (red), 4 level wavelets (blue) and 2 level wavelets (green) (**B**); PAA approximations of the same sequence with 8 (red), 5 (blue) and 3 (green) coefficients (**C**).

**Figure 2 viruses-11-00394-f002:**
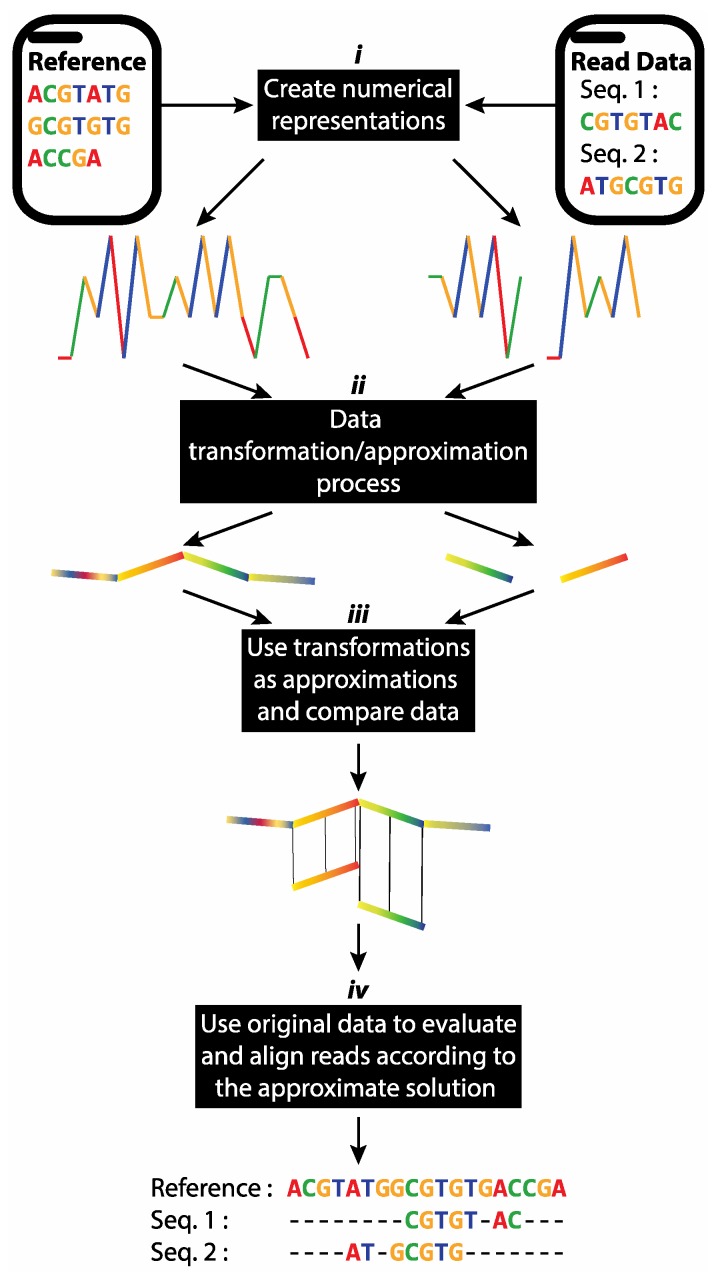
Overview of our proposed methodology using time series transformation/approximation methods: (***i***) Creation of numerical representations of input sequences. (***ii***) Application of an appropriate signal decomposition method to transform sequences into their feature space. (***iii***) Use of approximated transformations to perform rapid data analysis in lower dimensional space. (***iv***) Validation of inferences against original, full-resolution input sequences. In the case of reference-based alignment and taxonomic classification, approximated read transformations were compared with a reference sequence. In our de novo implementation, pairwise comparisons were performed between all of the approximated read transformations.

**Figure 3 viruses-11-00394-f003:**
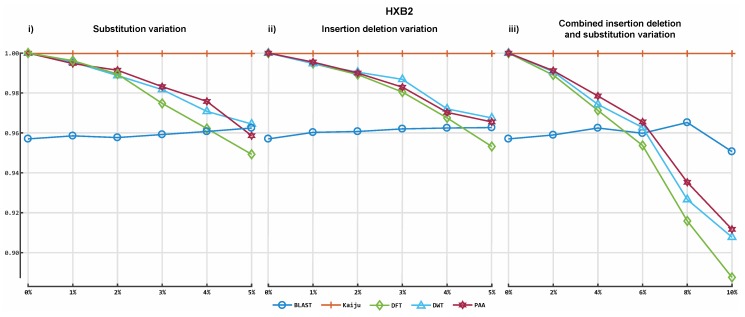
Accuracy of our prototype classification implementation and two established tools on HIV-1 HXB2 simulated datasets. All plots illustrate the *F*-measures obtained on the 16 different HIV datasets. The y-axis indicates the *F*-measure score, and the x-axis depicts the reads data files. Plot 3-i depicts the *F*-measures obtained for each classifier on the simulations with 0% to 5% of substitution variation rate. Plot 3-ii illustrates the *F*-measures obtained for each classifier on the simulations with 0% to 5% uniform insertion/deletion variation, and plot 3-iii illustrates the *F*-measures obtained for each tool on simulations of uniform 0% to 10% insertion/deletion and substitution variation.

**Figure 4 viruses-11-00394-f004:**
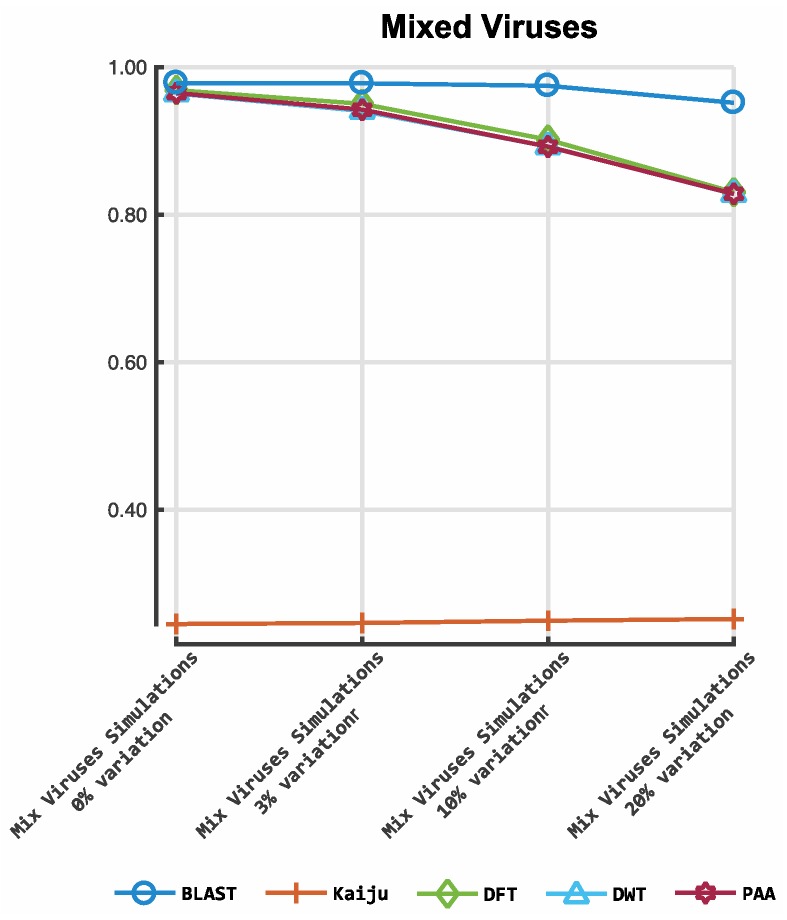
Accuracy of our prototype classification implementation and two established tools on mixed viruses simulated datasets. The *y*-axis indicates the *F*-measure score, and the *x*-axis depicts the reads data files. The plot depicts the *F*-measures obtained for each classifier on the mixed virus simulations. DFT: discrete Fourier transform; DWT: discrete wavelet transform; PAA: piece-wise aggregate approximation.

**Figure 5 viruses-11-00394-f005:**
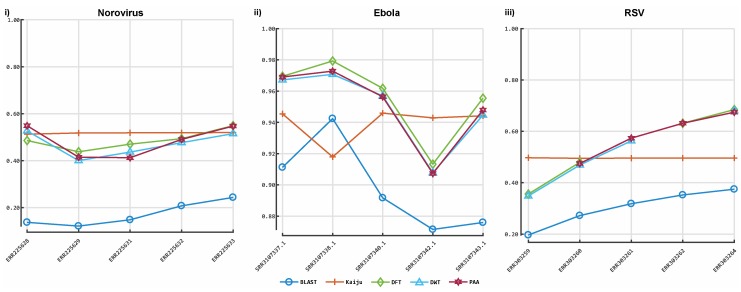
Accuracy of our prototype classification implementation and two established tools on real sequences. The *y*-axis indicates the *F*-measure score, and the *x*-axis depicts the reads data files. Plot 5-i depicts the *F*-measures obtained for each classifier on the Norovirus sequences data. Plot 5-ii illustrates the *F*-measures obtained for each classifier on the Ebola sequence data. Plot 5-iii illustrates the *F*-measures obtained for each tool on Respiratory syncytial virus (RSV) sequence data. DFT: discrete Fourier transform; DWT: discrete wavelet transform; PAA: piece-wise aggregate approximation.

**Figure 6 viruses-11-00394-f006:**
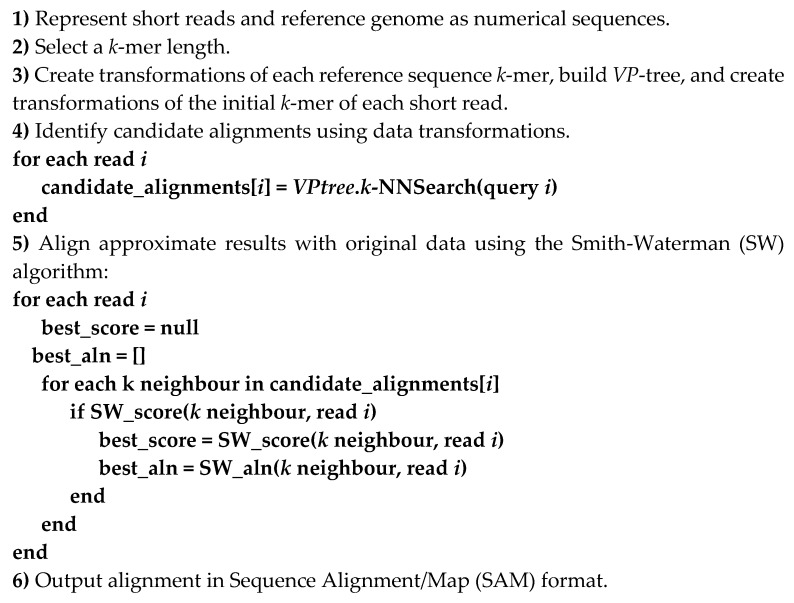
Pseudocode for the alignment procedure.

**Figure 7 viruses-11-00394-f007:**
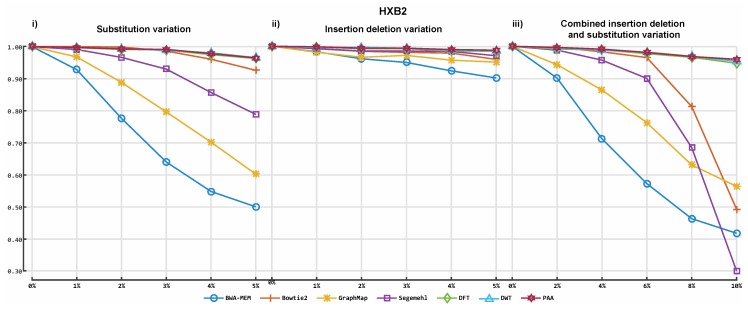
Accuracy of our prototype reference alignment implementation and four established tools on HIV-1 HXB2 simulated datasets. This Figure illustrates the *F*-measures obtained on the 16 different HIV datasets. Plot 6-(**i**) depicts the *F*-measures obtained for each aligner on the simulations with 0% to 5% of substitution variation rate. Plot 6-(**ii**) illustrates the *F*-measures obtained for each aligner on the simulations with 0% to 5% uniform insertion/deletion variation, and plot 6-(**iii**) illustrates the *F*-measures obtained for each tool on simulations of uniform 0% to 10% insertion/deletion and substitution variation. DFT: discrete Fourier transform; DWT: discrete wavelet transform; PAA: piece-wise aggregate approximation.

**Figure 8 viruses-11-00394-f008:**
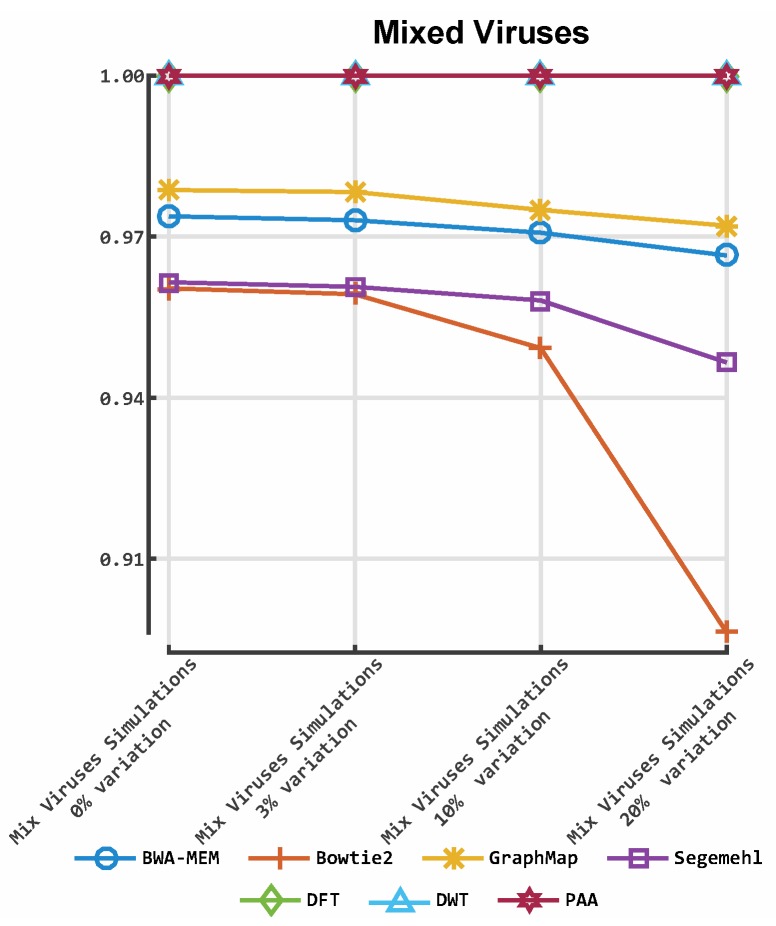
Accuracy of our prototype aligner implementation and four established tools on mixed viruses simulated datasets. The *y*-axis indicates the *F*-measure score, and the *x*-axis depicts the reads data files. The plot depicts the *F*-measures obtained for each aligner on the mixed virus simulations. DFT: discrete Fourier transform; DWT: discrete wavelet transform; PAA: piece-wise aggregate approximation.

**Figure 9 viruses-11-00394-f009:**
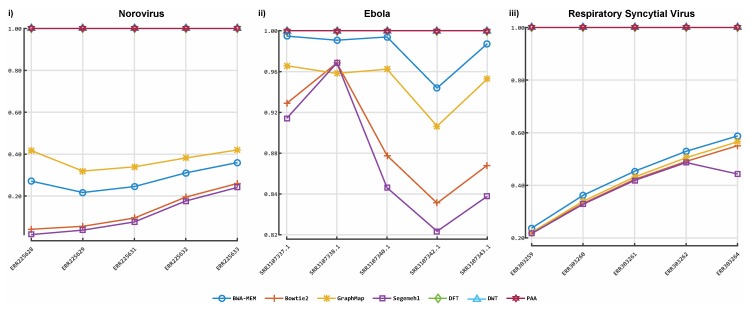
Accuracy of our prototype aligner implementation and four established tools on real sequences datasets. The *y*-axis indicates the *F*-measure score, and the *x*-axis depicts the reads data files. Plot 8-(**i**) depicts the *F*-measures obtained for each aligner on the Norovirus sequences data. Plot 8-(**ii**) illustrates the *F*-measures obtained for each aligner on the Ebola sequences data. Plot 8-(**iii**) illustrates the *F*-measures obtained for each tool on the Respiratory syncytial virus (RSV) sequences data. DFT: discrete Fourier transform; DWT: discrete wavelet transform; PAA: piece-wise aggregate approximation.

**Figure 10 viruses-11-00394-f010:**
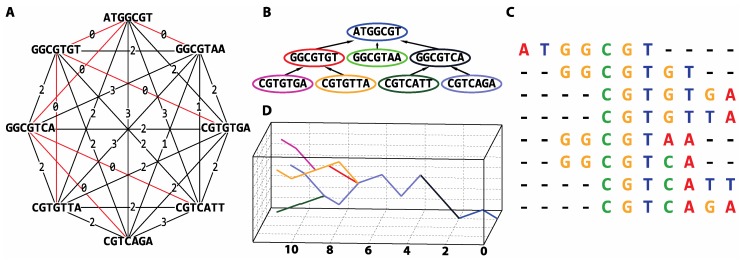
A de novo assembly methodology for numerically represented nucleotide reads. All-against-all sequence comparison (**A**) enables the construction of a read graph with weighted edges. The weight assigned to each edge is the smallest pairwise distance obtained between every possible *k*-mer representation of the two reads. In this example, a *5*-mer was used. The smallest distance between every possible *k*-mer can be obtained by either using a sliding window approach or break reads every possible subsequence with length *k.* (**B**) The shortest path in the graph is identified with a breadth-first search algorithm (red coloured edges) thereby (**C**) enabling read alignment. A DNA walk representation of the overlapped reads (**D**) may subsequently be used as a three-dimensional graphical portrayal of the reads, illustrating alignment characteristics.

**Figure 11 viruses-11-00394-f011:**
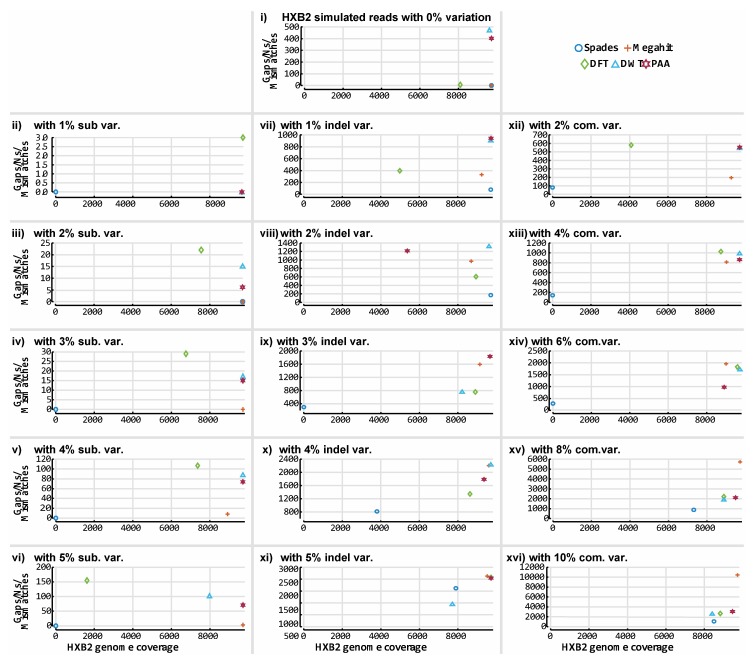
Accuracy of our prototype de novo assembly implementation and two established tools on HIV-1 HXB2 simulated datasets. The contigs obtained for each assembler were evaluated against the reference genome used to generate the simulated data. BLASTn was used to align all contigs to an HIV-1 HXB2 reference genome and determine genome coverage. The *y*-axis indicates the number of gaps and mismatches that exist in the contigs obtained for each tool, and the *x*-axis depicts the length of the genome the reported contigs cover. The contigs obtained from the assembly of the HIV-1 HXB2 simulated short read data were evaluated against the K03455 reference genome. Plot 10-**i** illustrates results obtained from all assemblers on variation-free data. Plots 10-**ii** to 10-**vi** illustrate results obtained from all assemblers on data with different levels of substitution variation. Plots 10-**vii** to 10-**xi** illustrate results obtained from all assemblers on data with different levels of insertion/deletion variation. Plots 10-**xii** to 10-**xvi** illustrate results obtained from all assemblers on data with different levels of combined insertion/deletion and substitution variation.

**Figure 12 viruses-11-00394-f012:**
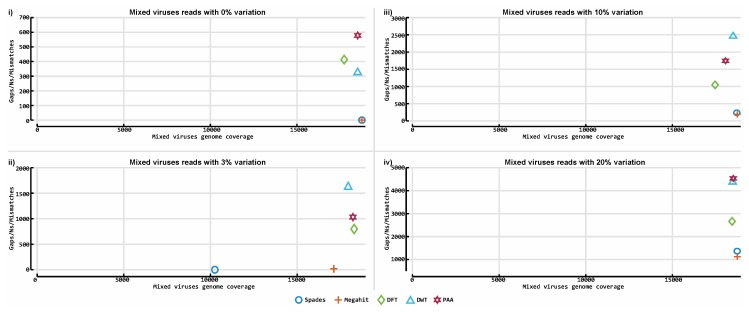
Accuracy of our prototype de novo assembly implementation and two established tools on mixed viruses simulated datasets. The contigs obtained for each assembler were evaluated against the reference genome that was used to generate the simulated data. BLASTn was used to align all contigs to an HIV-1 HXB2 reference genome and determine how much of the particular genome they cover. The *y*-axis indicates the number of gaps and mismatches that exist in the contigs obtained for each tool, and the *x*-axis depicts the length of the genome the reported contigs cover. The contigs obtained from the mixed virus simulated dataset were evaluated against the KM198529, KM198528, KM198511, KM198500, KM198486, KU296608, KU296553, KU296549, KU296528, KU296416, KP317952, KP317946, KP317934, KP317923 and KP317922 references genomes. Plots 11-**i** to 11-**iv** illustrate results obtained from all assemblers on data with 0%, 3%, 10% and 20% variation levels accordingly.

**Table 1 viruses-11-00394-t001:** Numerical nucleotide sequence representation methods.

Method	Numerical Representation
**Integer number**	A=1, C=−1, G=2, T=−2, N=0
**Real number**	A= −1.5, C=0.5,G= −0.5, T=1.5, N=0.0
**EIIP**	A=0.1260, C=0.1340, G=0.0806, T=0.1335, N= 0
**Atomic**	A=70, C=58, G=78, T=66, N= 0
**Pair**	A or T=1, C or G=−1, N=0
**Complex number**	A=1+1i, C= −1+1i, G=−1−1i, T=1−1i, N=0+0i
**DNA Walk**	A= [1,0], C= [0,1], G= [0,−1], T= [−1,0], N= [0,0]
**Tetrahedron**	A=[0,0,1], C=[−23,−63,13 ], G=[−23, −63, −13], T=[2×23, 0,−13], N=[0,0,0]
**Voss indicator**	A=[0,0,1,0], C=[1,0,0,0], G=[0,1,0,0], T=[0,0,0,1], N=[0,0,0,0]

**Table 2 viruses-11-00394-t002:** Simulated read data. Each row contains details for each simulated dataset (i.e., virus family, virus, GenBank ID, variation type, variation level, number of reads and simulator used to generate data). Abbreviations: Ins, insertions; Del, deletions and Sub, substitutions.

Family	Virus	GenBank Genome ID	Variation Type (%)			Reads	Simulator
			Ins	Del	Sub		
HIV	HXB2	K03455	0.0	0.0	0.0	2133	CuReSim
HIV	HXB2	K03455	0.0	0.0	1.0	2133	CuReSim
HIV	HXB2	K03455	0.0	0.0	2.0	2133	CuReSim
HIV	HXB2	K03455	0.0	0.0	3.0	2133	CuReSim
HIV	HXB2	K03455	0.0	0.0	4.0	2133	CuReSim
HIV	HXB2	K03455	0.0	0.0	5.0	2133	CuReSim
HIV	HXB2	K03455	0.5	0.5	0.0	2133	CuReSim
HIV	HXB2	K03455	1.0	1.0	0.0	2133	CuReSim
HIV	HXB2	K03455	1.5	1.5	0.0	2133	CuReSim
HIV	HXB2	K03455	2.0	2.0	0.0	2133	CuReSim
HIV	HXB2	K03455	2.5	2.5	0.0	2133	CuReSim
HIV	HXB2	K03455	0.5	0.5	1.0	2133	CuReSim
HIV	HXB2	K03455	1.0	1.0	2.0	2133	CuReSim
HIV	HXB2	K03455	1.5	1.5	3.0	2133	CuReSim
HIV	HXB2	K03455	2.0	2.0	4.0	2133	CuReSim
HIV	HXB2	K03455	2.5	2.5	5.0	2133	CuReSim
Mixed Viruses: Caliciviridae, Filoviridae, Pneumoviridae	Norovirus, Ebola virus, RSV	KM198529, KM198528, KM198511, KM198500, KM198486, KU296608, KU296553, KU296549, KU296528, KU296416, KP317952, KP317946, KP317934, KP317923, KP317922	0.0	0.0	0.0	200,000	WGSIM
Mixed Viruses: Caliciviridae, Filoviridae, Pneumoviridae	Norovirus, Ebola virus, RSV	KM198529, KM198528, KM198511, KM198500, KM198486, KU296608, KU296553, KU296549, KU296528, KU296416, KP317952, KP317946, KP317934, KP317923, KP317922	1.0	1.0	1.0	200,000	WGSIM
Mixed Viruses, Caliciviridae, Filoviridae, Pneumoviridae	Norovirus, Ebola virus, RSV	KM198529, KM198528, KM198511, KM198500, KM198486, KU296608, KU296553, KU296549, KU296528, KU296416, KP317952, KP317946, KP317934, KP317923, KP317922	3.33	3.33	3.33	100,000	WGSIM
Mixed Viruses, Caliciviridae, Filoviridae, Pneumoviridae	Norovirus, Ebola virus, RSV	KM198529, KM198528, KM198511, KM198500, KM198486, KU296608, KU296553, KU296549, KU296528, KU296416, KP317952, KP317946, KP317934, KP317923, KP317922	6.66	6.66	6.66	200,000	WGSIM

**Table 3 viruses-11-00394-t003:** Real short reads data. Rows contain information for each real reads’ dataset (i.e., virus family, virus, genome strain GenBank ID, SRA project ID, number of reads and technology used to sequence data). SRA: Sequence Read Archive; ENA: European Nucleotide Archive.

Family	Virus	Amplicon/Random Primer	GenBank Genome ID	ENA/SRA_ID	Reads	Sequencing Technology
Caliciviridae	Norovirus	Amplicon	KM198486	ERR225628	2126502	Illumina MiSeq
Caliciviridae	Norovirus	Amplicon	KM198500	ERR225629	3037674	Illumina MiSeq
Caliciviridae	Norovirus	Amplicon	KM198511	ERR225631	3285078	Illumina MiSeq
Caliciviridae	Norovirus	Amplicon	KM198528	ERR225632	4361884	Illumina MiSeq
Caliciviridae	Norovirus	Amplicon	KM198529	ERR225633	5187234	Illumina MiSeq
Filoviridae	Ebola virus	Amplicon	KU296608	SRR3107337	522968	Ion Torrent PGM
Filoviridae	Ebola virus	Amplicon	KU296549	SRR3107338	771031	Ion Torrent PGM
Filoviridae	Ebola virus	Amplicon	KU296416	SRR3107340	186657	Ion Torrent PGM
Filoviridae	Ebola virus	Amplicon	KU296553	SRR3107342	478346	Ion Torrent PGM
Filoviridae	Ebola virus	Amplicon	KU296528	SRR3107343	42410	Ion Torrent PGM
Pneumoviridae	RSV	Amplicon	KP317934	ERR303259	7275032	Illumina MiSeq
Pneumoviridae	RSV	Amplicon	KP317922	ERR303260	9278070	Illumina MiSeq
Pneumoviridae	RSV	Amplicon	KP317946	ERR303261	11111114	Illumina MiSeq
Pneumoviridae	RSV	Amplicon	KP317923	ERR303262	13293226	Illumina MiSeq
Pneumoviridae	RSV	Amplicon	KP317952	ERR303263	15237848	Illumina MiSeq

**Table 4 viruses-11-00394-t004:** Reference genomes used during classification and reference-based alignment.

Family	Virus	GenBank ID:	Length (nt)
Retroviridae	Human immunodeficiency virus 1 (HXB2)	K03455	9179
Caliciviridae	Norovirus	KM198509.1	7425
Filoviridae	Zaire ebolavirus	KM034562.1	18957
Pneumoviridae	Human orthopneumovirus (Respiratory Syncytial Virus)	KP317934.1	15233

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
