# Peer review of "The Utility of Data Transformation for Alignment, De Novo Assembly and Classification of Short Read Virus Sequences"

_viruses, 2019, doi:10.3390/v11050394_

Round 1
Reviewer 1 Report
While I find the approach and results of interest, it is difficult to know to what extent they are significant, robust, etc. As stated in the text, the methods of the authors can be parameterized, but we are not told what parameters were used. Similarly, a comparison is done against many other published methods but we are not told with what parameters they were run (or what parameter settings might have been more appropriate for them or why). Finally, the data sets used in all the experiments should be available. Any claim to perform better than a certain other algorithm should provide enough information for others to verify the claim and to experiment with parameter settings to establish whether the claim holds broadly or narrowly, under what circumstances, etc. None of this can be done given the data currently presented. Fortunately, this is easy to rectify. It would be good to have some idea of how robust the results in the paper are in this regard.
As a reviewer, it is very discouraging to receive a paper with so many syntactical, grammatical, and other simple errors in it that one can see immediately, even on just a cursory reading. At least two of the authors are native English speakers, and the paper contains a statement that all authors have read and approved the final manuscript. I very much doubt that to be the case. It's an unpleasant feeling to be spending one's Saturday finding basic errors in a submitted paper instead of spending time doing more pleasant things when it is clear that the authors put in only a small fraction as much care themselves. It sends a terrible message: "we didn't care enough to even read our own paper properly". How can anyone expect a reviewer put in any serious effort in such a context? It takes the thinking of the reviewer from "this might be interesting to review" to "wow, it looks like these people didn't even read their own paper!" See below for the many small problems along those lines.
Line 22-24: The sentence mixes present and past tense and also between singular and plural.
Line 44: Such approaches includes -> Such approaches include
Line 46: and graph-based methods -> graph-based methods
Line 65: "substantially great diversity". Use either substantial or great, not both.
Line 65: amount -> number
Line 72: data intensive fields -> data-intensive fields
Line 96: with same lengths -> with the same lengths
Line 105: The n2 should have the 2 as a superscript
Line 110: Change to "A phenomenon known as the ‘curse of dimensionality’"
Line 130 (eq. 1): What do the vertical lines indicate? They seem to add nothing and are confusing. The for all i in s_n is also confusing. Surely S_n is the nucleotide at position n in S. So it makes no sense to say for all i in this nucleotide. What's wanted instead, I imagine, is for 1 <= i <= n. This equation should be simplified - it's a weird excess of symbolism trying to formalize an extremely simple mapping that is already perfectly well explained in the final sentences of the previous paragraph. It looks like an amateurish attempt to add mathematical formalism.
Line 135: Voss indictor -> Voss indicator
Line 145: Should say "less than or equal"
Line 146: "guaranteeing against false negative results". It would be nice if you could add a sentence or two to explain what is meant here and under what circumstances there's a guarantee. I guess I could look up ref #43 but I'd rather have a summary from you.
Line 147: "and PAA approximation method" doesn't read properly. Should it be "and a PAA approximation method" or "and the PAA approximation method"?
Line 152: "transform (FFT) transform" would be easier to read if it said "transform (FFT) both transform"
Line 159/160: "from their original domain to their time-frequency, accommodating for 160 changes in signal frequency over time" should be "from their original domain to time-frequency, allowing for 160 changes in signal frequency over time" (not 2 suggested changes in there).
Line 171 has "VP-indexing tree" whereas line 174 has "VP-tree indexing"
Line 174: implemented in metric space -> implemented in a metric space?
Line 186: to visit either one or both of the nodes -> to visit either one or both of the child nodes?
Line 192: Why the past tense ("used")? The rest of the paragraph is in the present tense.
Line 208: "approach under every eventuality" This is too broad, without giving some detail. What does "every eventuality" include? If you know what that list would include, why leave it to chance and have a simulator construct the data set? I'd either say "under a wide range of input data" or else spell out what "every eventuality" includes and say why the simulator is guaranteed to spit out data that test each of these scenarios. A devil's advocate argument trivially undermines the current claim: a stochastic input data generator might not in fact generate test data covering "every" possible eventuality. Saying "a wide range" is fine, because your next paragraph goes on to say what the setup actually was.
Line 244/245: "Any hit on the correct section of the genome and in the correct direction report is considered a true positive result" would read better if it were "Any reported match on the correct section of the genome and in the correct direction is considered a true positive result"
Line 249: taking in consideration -> taking into consideration
Line 259: defines (VP-Tree) but this has already been used many times.
Line 267: "classification hits" I always think it's better to say "matches" instead of hits, for the benefit of non-native speakers.
Line 268: "The information generated for each classification includes"... It sounds like you have what you'd need to output SAM format. Could you do that? It might make the tool more generally useful for people doing downstream processing. Or maybe that's what you're doing already - in which case you should say so.
Line 283/284: "only following behind in on the high variation rates datasets" doesn't make sense.
Line 284: where classified -> were classified
Line 287: where classified -> were classified
Line 298: "well established, widely used, stated of the art" should be "well-established, widely used, state of the art" (note that there are 2 changes in that correction).
Line 302: "each aligner’s output an calculation" should be "and calculation"?
Line 311: "Figure 8-i to 8-iii" -> "Figures 8-i to 8-iii"
Line 320: "where reads where aligned" -> where reads were aligned
Line 325: ASBN should first be given in parentheses on line 321.
Line 341: in terms of size and nucleotide sequences -> in terms of size and nucleotide sequence
Line 351: "experiment(Figure 10" needs a space.
Line 393: "representations representa a tractable" represents? Maybe "offers a tractable"?
Line 404: "BC, MP and MC help on" should be "helped"
Figure 1: legend (on right) should say "1 frequency" (not 1 frequencies).
Figure 1: instead of repeating "numerical transformation" three times in the legends on the right, why not just say it once in the main legend to the figure?
Figure 3: legend says "Plot 6-ii illustrates" but that should be Plot 3-ii.
Figure 5: The legend contains repeated text: "The Y axis indicates the F-measure score, and the X axis depicts the reads data files. The Y axis indicates the F-measure score, and the X axis depicts the reads data files."
Figure 8: Also contains duplicated text: "The Y axis indicates the F-measure score, and the X axis depicts the reads data files. The Y axis indicates the F-measure score, and the X axis depicts the reads data files."
Figure 9A: Why are the edges labeled zero red, while the others are black? I don't understand the distances. Why is the distance between GGCGTAA and GGCGTCA 3 despite them having only one nucleotide that differs, while ATGGCGT and GGCGTAA differ at 6 out of 7 locations but have a distance of zero.
Figure 10 contains the text "simualted" in the title of all sub-plots
Figure 10 could be de-cluttered by having plots share X axis labels and values and Y axis labels.
Figure 10 says "Plots 10-xiii to 10-xvi illustrate" but this should be "Plots 10-xii to 10-xvi illustrate".
Figure 10 legend (line 488): "on variation free data" -> "on variation-free data"
Figure 11 legend: "how much of the particular genome they do cover" should be "how much of the genome they cover".
Table 1: The nucleotide letter is sometimes in italics, sometimes not.
Author Response
Reviewer 1 response:
We would like to thank reviewer 1 for the constructive comments and for taking the time to individually address a number of typographical errors featuring in the originally submitted manuscript. We have addressed these mistakes and further clarified a number of points raised by the reviewer. We have carefully proofread the manuscript.
Comment 1:
While I find the approach and results of interest, it is difficult to know to what extent they are significant, robust, etc. As stated in the text, the methods of the authors can be parameterized, but we are not told what parameters were used. Similarly, a comparison is done against many other published methods but we are not told with what parameters they were run (or what parameter settings might have been more appropriate for them or why). Finally, the data sets used in all the experiments should be available. Any claim to perform better than a certain other algorithm should provide enough information for others to verify the claim and to experiment with parameter settings to establish whether the claim holds broadly or narrowly, under what circumstances, etc. None of this can be done given the data currently presented. Fortunately, this is easy to rectify. It would be good to have some idea of how robust the results in the paper are in this regard.
Response:
We have revised the document to clarify the parameters used for each of the tools in our analysis. Concerning the statement ‘Finally, the data sets used in all the experiments should be available’, all of the short read data used in this study are either computationally generated (using specified parameters and freely available software, and available through https://github.com/Avramis/Supporting-data) or available in the ENA public database with accession numbers listed in the Table 3, columns named “GeBank genome ID”, “ENA/SRA_ID”, accompanied by the sequencing technology and total reads count. The study specifically used short read data from known viral sequencing projects, for which accession numbers have been included and relevant publications cited. We have added explicit instructions to point readers towards the original datasets used in the study.
Comment 2:
Line 130 (eq. 1): What do the vertical lines indicate? They seem to add nothing and are confusing. The for all i in s_n is also confusing. Surely S_n is the nucleotide at position n in S. So it makes no sense to say for all i in this nucleotide. What's wanted instead, I imagine, is for 1 <= i <= n. This equation should be simplified - it's a weird excess of symbolism trying to formalize an extremely simple mapping that is already perfectly well explained in the final sentences of the previous paragraph. It looks like an amateurish attempt to add mathematical formalism.
Response:
A revised equation has been provided.
Comment 3:
Line 146: ‘guaranteeing against false negative results’. It would be nice if you could add a sentence or two to explain what is meant here and under what circumstances there's a guarantee. I guess I could look up ref #43 but I'd rather have a summary from you.
Response:
A detailed explanation has been provided for the phrase ‘guaranteeing against false negative result’, in order to make it more explicit for the reader and to save the reviewer from having to read the supporting literature. The sentence itself has also been changed to read ‘precluding false negative results’.
Comment 4:
Line 268: "The information generated for each classification includes"... It sounds like you have what you'd need to output SAM format. Could you do that? It might make the tool more generally useful for people doing downstream processing. Or maybe that's what you're doing already - in which case you should say so.
Response:
Our tool as currently designed generates a verbose BLAST-like alignment together with a summary TSV file.
The former comprehensively reports the generated alignment for each read
e.g.
Genome name
Range 1: 745 to 1121
Query 1 AAATTTCTCCTACTGGGATAGGTGGATTATTTGTCATCCATCCTATTTGTTCCTGAAGGG 60
||||||||||||||||||||||||||||||||||||||||||||||||||||||||||||
Sbjct 1121 AAATTTCTCCTACTGGGATAGGTGGATTATTTGTCATCCATCCTATTTGTTCCTGAAGGG 1062
Query 61 TACTAGTAGTTCCTGCTATGTCACTTCCCCTTGGTTCTCTCATCTGGCCTGGTGCAATAG 120
||||||||||||||||||||||||||||||||||||||||||||||||||||||||||||
Sbjct 1061 TACTAGTAGTTCCTGCTATGTCACTTCCCCTTGGTTCTCTCATCTGGCCTGGTGCAATAG 1002
Query 121 GCCCTGCATGCACTGGATGCACTCTATCCCATTCTGCAGCTTCCTCATTGATGGTCTCTT 180
||||||||||||||||||||||||||||||||||||||||||||||||||||||||||||
Sbjct 1001 GCCCTGCATGCACTGGATGCACTCTATCCCATTCTGCAGCTTCCTCATTGATGGTCTCTT 942
Query 181 TTAACATTTGCATGGCTGCTTGATGTCCCCCCACTGTGTTTAGCATGGTGTTTAAATCTT 240
||||||||||||||||||||||||||||||||||||||||||||||||||||||||||||
Sbjct 941 TTAACATTTGCATGGCTGCTTGATGTCCCCCCACTGTGTTTAGCATGGTGTTTAAATCTT 882
Query 241 GTGGGGTGGCTCCTTCTGATAATGCTGAAAACATGGGTATCACTTCTGGGCTGAAAGCCT 300
||||||||||||||||||||||||||||||||||||||||||||||||||||||||||||
Sbjct 881 GTGGGGTGGCTCCTTCTGATAATGCTGAAAACATGGGTATCACTTCTGGGCTGAAAGCCT 822
Query 301 TCTCTTCTACTACTTTTACCCATGCATTTAAAGTTCTAGGTGATATGGCCTGATGTACCA 360
||||||||||||||||||||||||||||||||||||||||||||||||||||||||||||
Sbjct 821 TCTCTTCTACTACTTTTACCCATGCATTTAAAGTTCTAGGTGATATGGCCTGATGTACCA 762
Query 361 TTTGCCCCTGGATGTTC 377
|||||||||||||||||
Sbjct 761 TTTGCCCCTGGATGTTC 745
Each line of the TSV file contains the following information:
- Read ID
- Number of times matched to a genome
- Highest score obtained from matching read to a given genome.
- Name of the genome with the highest matching score
- Direction in which the read aligned to the genome
- Starting position of the match
While we do agree that SAM output could be a useful enhancement for some users, it is not standard practice for a classifier to generate SAM output, and we feel that generating alignment summaries in a tabular format easily human and machine readable is sufficient to support our findings, for what is a prototype classification approach. For an aligner we agree that SAM output is very valuable, hence its implementation in our ALBN approach.
Comment 5:
Figure 1: instead of repeating "numerical transformation" three times in the legends on the right, why not just say it once in the main legend to the figure?
Response:
We thank the reviewer for this suggestion and we have revised the Figure 1 by adding an abbreviation for numerical transformation for the second two instances.
Comment 6:
Figure 10 could be de-cluttered by having plots share X axis labels and values and Y axis labels.
Response:
We thank the reviewer for suggesting this improvement, we have revised Figure 10, removing the redundant labelling.
Reviewer 2 Report
Tapinos and colleagues show the usefulness of data transformation and reduction in the context of genomics to deal with alignment, de novo assembly and classification. Authors employ simulated data and real data. In my opinion, the manuscript is very well written, the figures are appropriated and well presented. The idea is quite original and sounds interesting!. I would accept the paper with minor changes. I have a couple of suggestions that I like authors address in the reviewed version of the manuscript:
1) Next generation sequencing is rapidly evolving and is moving to long-read sequencing (e.g Nanopore technology) which nowadays is providing great data and likely in the coming 2-4 years even the technology will be improved. With long-read sequencing data, intense computation calculation is clearly reduced. I would like that authors address this issue and put in the context how useful would be to apply transformation and reduction of data (as used in this manuscript) in a scenario in which long read dominates the scene of genomic and metagenomics. Maybe is not needed at all?
2) The introduction addresses several aspects about how intense and complicated is the calculation of large alignments and assemblies (time consuming, high informatics resources, etc...). However, actually I miss in the result and discussion sections, a paragraph discussing and showing how much we (the research community) "save" in terms of resources, time, computing, etc...when we use data transformation. Because, if we do not save in any of these aspects, then why transforming the data? then, better using the actual data. I think the reader would appreciate it. I´m totally convinced that the accuracy is very nice with the method presented by authors, and is very robust. However, for the virology community, it would be good to show some results and data to convince them about the benefits of using data transformation and compression. Let´s say that I have a dataset to assembly with SPAdes (my preferred assembler). After doing the same with the methdology proposed by the authors, how much in terms of time, resources I save...??
Again, i think authors have done a very good job!
Author Response
Reviewer 2 response:
We would like to take the opportunity to thank reviewer 2 for the useful comments and suggestions.
Comment 1:
Next generation sequencing is rapidly evolving and is moving to long-read sequencing (e.g Nanopore technology) which nowadays is providing great data and likely in the coming 2-4 years even the technology will be improved. With long-read sequencing data, intense computation calculation is clearly reduced. I would like that authors address this issue and put in the context how useful would be to apply transformation and reduction of data (as used in this manuscript) in a scenario in which long read dominates the scene of genomic and metagenomics. Maybe is not needed at all?
Response:
We agree that this is an important point, and have added a paragraph to the discussion section demonstrating how and why the use of data transformations will be relevant for analysing long reads generated by, for example, The Oxford Nanopore platform.
Comment 2:
The introduction addresses several aspects about how intense and complicated is the calculation of large alignments and assemblies (time consuming, high informatics resources, etc...). However, actually I miss in the result and discussion sections, a paragraph discussing and showing how much we (the research community) "save" in terms of resources, time, Thanks so much!!! computing, etc...when we use data transformation. Because, if we do not save in any of these aspects, then why transforming the data? then, better using the actual data. I think the reader would appreciate it. I´m totally convinced that the accuracy is very nice with the method presented by authors, and is very robust. However, for the virology community, it would be good to show some results and data to convince them about the benefits of using data transformation and compression. Let´s say that I have a dataset to assembly with SPAdes (my preferred assembler). After doing the same with the methodology proposed by the authors, how much in terms of time, resources I save...??
Response:
As we mentioned in our manuscript our tools are prototype implementations (especially ASBN), with an emphasis on the accuracy of the analysis, and cannot be directly compared to existing tools, thus we did not comment on that aspect. In our research we want to demonstrate the application of signal processing techniques to nucleotide sequences. We stated the theoretical time complexity for the tools that we are using in the manuscript (and our analysis can be performed in O(nlog(n)) time). However, in terms of timing, CBN and ALBN tools had a reasonable execution time, similar to KAIJU and Bowtie2 accordingly. We believe that specific characteristics of long read sequences – namely their length and homopolymeric indel rates make them particularly amenable to analysis using approaches conceived for analysing time series that we introduce in this study. Long reads are afflicted to a greater extent than short reads by the curse of dimensionality. Under the curse of dimensionality, typically the contrast on the distances between pairs of data points is small, thus the concept of proximity becomes meaningless from a qualitative perspective. The data transformations we discuss in this study cope with this problem well, and we believe that the methodology has lots of potential for performance improvements in the long read domain.